# A Glimmer of Hope for Patients with a T3 Transformation Zone: miRNAs Are Potential Biomarkers for Cervical Dysplasia

**DOI:** 10.3390/diagnostics13243599

**Published:** 2023-12-05

**Authors:** Julia Wittenborn, Eva-Marie Flasshove, Tomas Kupec, Laila Najjari, Elmar Stickeler, Jochen Maurer

**Affiliations:** Department of Obstetrics and Gynecology, University Hospital of the RWTH Aachen, 52074 Aachen, Germanyjmaurer@ukaachen.de (J.M.)

**Keywords:** human papilloma virus (HPV), colposcopy, cervical cancer screening

## Abstract

Background: This pilot study assesses the potential use of miRNAs in the triage of colposcopy patients with type 3 (nonvisible) cervical transformation zone (TZ). Type 3 TZ is a constitutional finding associated with many problems and controversies in colposcopy patient management. Here, we present miRNAs as a potential biomarker for the detection of CIN3 in these cases. Materials and methods: Cervical mucosa samples (CMS) were collected from patients presenting with T3 transformation zone during routine workup using the Cytobrush. Depending on the histological and cytological result, as well as the result of the routinely performed HPV PCR, patients were divided into three groups: patients with a high-grade intraepithelial lesion (CIN3) and a positive high-risk HPV test (CIN3 group), patients without an intraepithelial lesion and a positive high-risk HPV test (HPV group), and healthy controls (N = no intraepithelial lesion and negative HPV test). The cervical mucus samples included in the study were tested for their expression levels of distinct miRNAs using qPCR. Results: All investigated miRNAs were consistently detectable in every sample. The CMSs of histologically graded CIN 3 showed consistently high expression levels of all eight miRNAs, whereas the CMSs from healthy patients (N) show generally lower expression levels. However, CMSs from patients of the HPV group represented a very heterogeneous group. Conclusions: The data presented here can provide a solid basis for future research into a triage test for patients with a T3 transformation zone on the basis of commonly used clinical equipment.

## 1. Introduction

Cervical cancer is the fourth most common malignancy in women worldwide [1], and the fourth most common cause of death from cancer in women in the world. There were 528,000 new cases of cervical cancer worldwide in 2012, resulting in 266,000 deaths, accounting for 7.5% of all female cancer deaths [1,2]. Squamous cell carcinomas account for over 80% of cervical cancer cases and are typically preceded by a premalignant condition called cervical intraepithelial neoplasia (CIN). CIN is classified into different grades, ranging from CIN 1 to CIN 3. In the Betheda classification system, CIN1 is defined as LSIL (low-grade squamous intraepithelial lesion) and CIN2 and CIN3 as HSIL (high-grade squamous intraepithelial lesion) based on histological characteristics, such as cellular differentiation, maturation, stratification, and nuclear abnormalities and mitoses. In addition, the proportion of the thickness of the epithelium and differentiated cells is used for grading CIN [3]. Nearly all cervical cancers are associated with high-risk HPV (human papillomavirus) infections. In many countries, screening for HPV has been added to the classical Pap smear in screening programs [4,5,6]. In Germany, a newly organized screening, based on cytology (PAP smear) and HPV testing, was introduced in 2020, replacing the previous cytology-only based program. Although high-risk HPV DNA testing has a high sensitivity, it remains difficult to detect clinically significant HPV infections that have the potential to transform and lead to health complications. [7]. Transforming HPV infections differs from productive infections in the way that the normal viral life cycle of HPV is disrupted, followed by overexpression of the viral oncogenes E6 and E7 [8]. The German screening guidelines for abnormal findings [9] demand a colposcopy in cases of suspicious cytology and positive high-risk HPV testing and also in cases of high-risk HPV persistence over one year with normal cytology [10]. As HPV infections occur at the transformation zone (TZ) of the cervix, which is the squamocolumnar junction between the endo- and ectocervix, the colposcopic examination focuses on this area. Depending on the visibility of the squamocolumnar junction, the transformation zone (TZ) is categorized into three types: type 1 is completely visible, type 2 is completely visible after splaying of the cervix, and type 3 (T3) is not completely visible [11]. TZ T3 is a constitutional finding, predominantly determined by hormones, obstetric history, and the lack of estrogens, representing a common situation during menopause. TZ T3 is related to controversies in colposcopy patient management, requiring skilled physician input and often a multidisciplinary approach. With the ageing population globally, it is anticipated to represent a growing issue in the forthcoming decades. As in the case of a type 3 transformation zone (TZ), only limited assessment of the transformation zone is possible; this may result in an increased risk of missing disease by up to twofold [12,13,14]. To improve the detection and reduce the risk of missing disease in this group of women, endocervical curettage (ECC) can be performed but remains controversial [15,16]. To ensure accurate diagnostics and treatment in women with incomplete visualization of the TZ, some guidelines recommend that a diagnostic loop electrosurgical excision procedure (LEEP) should be considered [9,17,18]. On the other hand, the indication for a diagnostic LEEP should be highly restricted, as it entails an invasive surgical procedure for the patient under general anaesthesia, which carries specific surgical risks like bleeding and postoperative cervical stenosis [19].

With the introduction of the new cervical cancer screening program in Germany in January 2020, colposcopies have become even more frequent and are a mandatory examination even for patients with one-time cervical abnormalities in case they carry an HPV high-risk infection [9]. Little attention has been paid to the psychological effects of these investigations. It has been shown before that patients being transferred for a colposcopic examination have high anxiety levels, and the prospect of a potentially painful examination is a key problem [20].

Keeping all this in mind, the search for effective and accurate minimally invasive biomarkers with high diagnostic and prognostic value, requiring minimal time and material costs, remains relevant.

Several biomarkers, including p16, ki-67, methylation, and genotyping, have been tested so far, but none were able to close the diagnostic gap in clinical practice [21,22]. On the other hand, miRNAs have become an interesting tool in distinguishing genetically different cellular entities and are a source of potential biomarkers [23,24,25,26]. MicroRNAs are noncoding regulatory RNAs 19–25 nucleotides (nt) in size that are produced by RNA polymerase II and II derived from transcripts of coding or noncoding genes. Many miRNAs are tissue specific or differentiation specific, and their temporal and lived expressions modulate gene expression at the post-transcriptional level by base pairing with complementary nucleotide sequences of target mRNAs. Actions of miRNAs exert profound effects on gene expression in almost every biological process, and aberrant miRNA expression is well recognized as a marker for several carcinomas [27,28].

Aberrant expression of miRNAs in cervical cancer and its precursor lesions have been studied before and have shown potential in distinguishing even the different CIN types [29]. In the present study, we sought to investigate the differential expression of miRNAs in patients with a T3 transformation zone, because the greatest need for biomarkers is in this large and continuously growing patient collective.

## 2. Materials and Methods

Cervical mucosa samples (CMS) were collected in our certified center for dysplasia at the university hospital in Aachen. Samples were collected from all patients presenting with a T3 transformation zone from 23 July 2021 to 21 December 2021. A total of 537 patients were screened for the presence of a T3 transformation zone; 167 specimens were collected, and 63 were included in the study (see Figure 1). The standardized colposcopic examination was performed in the DKG-certified colposcopy unit of University Hospital Aachen by experienced and highly qualified AG CPC-certified personnel. It was executed with a Leisegang 3MCV colposcope. Every colposcopy included the systematic collection of a conventional PAP smear (control cytology). All patients received HPV PCR testing and a colposcopic examination. Colposcopy-directed biopsies were taken as well as an endocervical curettage [30]. Depending on the histological and (control-)cytological result, as well as the result of the performed HPV PCR, patients were divided into three groups: patients with a high-grade intraepithelial lesion (CIN3) and a positive high-risk HPV test (CIN3 group), patients without an intraepithelial lesion and a positive high-risk HPV test (HPV group), and healthy controls (N = no intraepithelial lesion and negative HPV test). Patients with no evidence of CIN 3 in histological sampling or excisional procedures were excluded from the study. Other exclusion criteria for the study included ongoing systemic hormonal treatments, individuals younger than 18 years old, and individuals older than 75 years old, present sexually transmitted diseases such as Chlamydia infection and other urogenital infections at the time of investigation. In case of a discrepancy between the cytological grade of dysplasia and the histological result, or unspecific pathological results due to insufficient cervical harvest from the biopsy, the patient sample was also excluded (see Figure 1).

Patient characteristics of the groups are displayed in Table 1.

The cervical mucus samples were obtained using the ‘Cytobrush Standard’, a swab of the cervical canal. The brushes were rotated for a full 360° to ensure complete coverage and obtain a comparable amount of sample. Following this, the brushes were streaked onto a specimen slide for cytological evaluation.

The remaining mucus and cells obtained from the cervical canal were promptly processed using the following procedure. The general assessment was carried out in accordance with the 2011 International Federation for Cervical Pathology and Colposcopy (IFCPC) colposcopic terminology for the cervix; transformation zone types were classified accordingly as 1, 2, or 3 [11].

After obtaining the cervical mucus samples and performing cytological evaluation, 5% acetic acid was applied to the cervical surface for further assessment. The colposcopic findings were then graded according to the International Federation for Cervical Pathology and Colposcopy (IFCPC) nomenclature.

The grading system included the following categories:“Normal findings”: this category encompassed findings such as polyps, viral warts, or metaplasia;“Minor changes”: thin, acetowhite epithelium, an irregular geographic border, a fine mosaic, and a fine punctation fell into this category;“Major changes”: dense acetowhite epithelium, rapid appearance of acetowhitening, cuffed crypt (gland) openings, a coarse mosaic, coarse punctuation, a sharp border, an inner border sign, and a ridge sign were classified as major changes;“Suspicious of invasion/cancer”: Findings falling into this category were suggestive of potential invasion or cancer.

The results of the colposcopic examination were then displayed in Table 2. A colposcopy-directed biopsy was taken in case of a visible acetowhite lesion. In some patients with multifocal lesions, more than one biopsy was taken from areas colposcopically identified as abnormal or doubtful. A biopsy was obtained from the area of the lesion with the worst features and closest to the squamocolumnar junction. In all cases, an endocervical curettage was performed because of the T3 transformation zone. Except for 4 patients in the control group who presented to our colposcopy unit for vulvar diseases, all patients included in the study had a histological workup of the cervix in order to most accurately place the patients in the right groups. The indication for surgical therapy, loop electrosurgical excision procedure (LEEP), or total laparoscopic hysterectomy (TLH) was set according to the German S3 guideline for the prevention of cervical cancer [9].

During the whole period of investigation, the Seegene Anyplex II HPV 28 (Seegene Germany GmbH-Merowingerplatz 1, Düsseldorf, Germany) detection kit was used. It simultaneously detects 19 high-risk and intermediate-risk HPV genotypes and 9 low-risk types. The results of HPV testing are displayed in Table 1. All patients received HPV testing.

The study was covered and enabled by the Centralized Biomaterial Bank at RWTH Aachen. The protocol used in the study received approval from the independent ethics committee of the faculty of medicine at RWTH (with ethics vote EK 206/09). Prior to participation in the study, written informed consent was obtained from each patient.

Alongside the routine diagnostic procedures such as cytology, histology, and HPV screening, the cervical mucus samples included in the study underwent testing to determine the expression levels of specific miRNAs. This additional analysis aimed to provide further insights into the potential role of miRNAs in cervical health and disease.

### 2.1. Processing and Lysis

To extract the remaining cervical mucus and cells from the brush, QIAzol Lysis Reagent (#79306, QIAGEN, Venlo, Netherlands) was used for lysis. The samples were incubated at room temperature for 10 min using an Eppendorf Thermo Mixer C5 (#5382000015, Eppendorf, Hamburg, Germany). These steps were carried out following the manufacturer’s protocol, ensuring that the necessary instructions were followed precisely for accurate and reliable results.

### 2.2. RNA Isolation, Reverse Transcription, and Amplification

RNA isolation and cleanup were carried out using the RNeasy MinElute Cleanup Kit by Qiagen (#74204). For this step, phenol-chloroform extraction was employed. Following extraction, all RNA samples were stored at −80 °C to maintain their stability.

To generate complementary DNA (cDNA) for the miRNA analysis, the TaqMan™ advanced cDNA Synthesis Kit (#A28007, Applied Biosystems™, Waltham, MA, USA) was utilized. As an exogenous control, the synthetic ath-mir-159a was integrated as a template during cDNA synthesis. This control was included to ensure the accuracy of the amplification process and to create a reliable qPCR construct. After preamplification, the cDNA samples were stored at −20 °C until they were ready for subsequent PCR analysis.

### 2.3. Real-Time qPCR

To determine the miRNA expression levels, a quantitative real-time PCR was performed using the LightCycler 480 Instrument II (#05015243001, Roche, Basel, Switzerland). The TaqMan^®^ MicroRNA Assays (#4427975, Applied Biosystems^®^) and TaqMan™ Fast Advanced Master Mix (#4444557, Applied Biosystems™) were used for this analysis.

In each reaction volume of 10 µL, 2.5 µL of diluted miRNA cDNA were included. To evaluate the expression values, the relative quantification method was applied. This involved calculating dCT, which refers to the difference in the crossing point (the number of cycles at which the fluorescence exceeds the threshold) values. These dCT values were then normalized using the corresponding values of control miRNA (ath-mir-159a) for each sample.

To ensure reliable results, three technical replicates were performed for each cervical mucus sample.

### 2.4. Statistical Analysis

Statistical analysis was conducted using Microsoft Excel and Graph Pad Prism 8.0 software. To evaluate significant differences between the sample groups, Mann–Whitney U-tests were performed.

## 3. Results

The final patient cohorts consisted of 19 healthy individuals in the control group (N), 23 patients with cervical intraepithelial neoplasia grade 3 (CIN3), and 21 patients with high-risk HPV infection and normal histology and control cytology (HPV). In the CIN3 group, the majority of patients had high-grade cytological abnormalities: 39.1% had PAP IVap (HSIL) and 26.1% had PAP IIID2 (HSIL). All patients in this group had a histologically confirmed CIN3. In 16 patients, colposcopic-led biopsy or endocervical curettage revealed CIN3, and, in 7 patients, it was found in the later-performed loop excisional procedure of the cervix (LEEP).

Based on the extensive literature research and personal experiences in the detection of miRNAs in cervical mucus and other body fluids, we selected eight miRNAs to be tested on patient CMS [23,29]. Hsa-miR-26b-5p, hsa-miR-142-3p, hsa-miR-143-3p, hsa-miR-191-5p, hsa-miR-223-3p, hsa-miR-338-3p, hsa-miR-205-5p, and hsa-miR-130a-3p were subsequently analyzed in a total of 63 patients.

All investigated miRNAs were consistently detectable in every sample. The expression levels were generally higher in patients of the CIN3 group than in the normal controls (N) (see Figure 2). The CMS of histologically graded CIN 3 showed consistently high expression levels of all eight miRNAs, whereas the CMSs from healthy patients (N) show generally only lower expression levels. The corresponding *p* values are displayed in Table 3 and Table 4. However, CMSs from patients of the HPV group represented a very heterogeneous group, including samples with higher expression of all of the eight tested miRNAs but also unchanged levels compared with healthy controls (see Figure 3). Looking at miRNA expression changes between patients with CIN 3 (HSIL) and high-risk HPV infection (CIN3 group), patients with high-risk HPV infection without intraepithelial neoplasia (HPV group), and healthy individuals (N) for expression levels of the investigated miRNAs divided by the SEM (standard error of the mean), one can see the described tendencies towards higher expression levels in the CIN3 group than in normal controls, in relation to the heterogeneous results of the HPV group (see Figure 4).

## 4. Discussion

Due to overwhelming evidence from long-term prospective cohorts and randomized clinical trials demonstrating that high-risk HPV DNA testing is considerably more sensitive than cervical cytology for the detection of cervical intraepithelial neoplasia grade and cancer, a newly organized screening based on cytology (PAP smear) and HPV testing was introduced in 2020 in Germany, replacing the previous cytology-only based program [6,31,32,33,34]. Despite the high sensitivity of high-risk HPV DNA testing, it remains challenging to identify clinically relevant, i.e., transforming, HPV infections [7]. Transforming HPV infections differ from productive infections in the way that the normal viral life cycle of HPV is halted, followed by overexpression of the viral oncogenes E6 and E7 [8]. The German screening guidelines for abnormal findings [9] demand a colposcopy in cases of suspicious cytology, positive high-risk HPV testing, and in cases of HPV persistence over one year with normal cytology [10]. Therefore, a new, rather large, patient collective in the German cervical cancer screening has led to a high influx of patients in all centres for colposcopy. This anticipated elevated colposcopy referral rate, after the introduction of HPV screening on the one hand and the reduced sensitivity rates of colposcopy in patients with a T3 transformation zone on the other hand, poses a major controversy [13,14,35]. Endocervical curettage is often uncomfortable or painful for the patients, and the rates of additional CIN2+ diagnosis are only between 0.8 and 6% [36]. Diagnostic LEEP as a surgical procedure is associated with high costs and morbidity for the patients [37]. In order to determine the presence of cervical intraepithelial neoplasia in this patient collective, biomarkers are desperately needed. Several biomarkers, like partial genotyping, p16/Ki-67, methylation, and others, have been tried before, but none of them has been proven helpful in clinical routine [22,38]. Thus, the search for effective and accurate minimally invasive biomarkers with high diagnostic and prognostic value, requiring minimal time and material costs, remains relevant. Here, we present miRNAs as a potential biomarker for the detection of CIN3 in cases of the T3 transformation zone.

All tested miRNAs had higher expression levels in patients with CIN3 than in the healthy controls.

One of the major difficulties of this study was the assignment of patients to the different study groups in light of the low sensitivity rates of all diagnostic measures at hand (colposcopy and endocervical curettage) Thus, we had to rely on cytology and histology when assigning the patients to the different groups. Unfortunately, we cannot be 100% sure if the patients in our healthy control group were actually completely healthy. Eleven patients in the control group were initially transferred to our department because of suspicious cytologies, but they had normal cytological results in our department (normal control cytologies). For the investigated patient collective of patients with a T3 transformation zone, we performed the most thorough workup possible, including HPV PCR testing, colposcopy-directed biopsies, and endocervical curettage. Endocervical curettage has a good sensitivity of 70–81% [39,40] in detecting CIN2+. Nonetheless, the rate of CIN2+ in women with HPV persistence and normal cytology is described to be up to 15% in diagnostic loop excisions in this collective [41].

In our own preliminary examination, we were able to show a differential expression of hsa-miR-26b-5p, hsa-miR-142-3p, hsa-miR-143-3p, hsa-miR-191-5p, hsa-miR-223-3p, and hsa-miR-338-3p for different types of cervical intraepithelial neoplasia. As we tested the expression levels in a patient collective irrespective of the transformation zone, we hypothesized that the detection rate would be even better in a collective of patients with a T3 transformation zone, as the Cytobrush will be in contact with the transformation zone with a much higher probability than in a mixed collective. The focus of the current study was to differentiate healthy individuals from patients with clinically relevant pathology rather than showing a differential expression of different CIN, as this matter is of much higher clinical relevance. A secondary goal was to differentiate transforming (clinically relevant) HPV infections from productive HPV infections. Therefore, the HPV group was implemented with seemingly clinical nonrelevant HPV detection.

Additionally, we investigated the expression of hsa-miR-205-5p and hsa-miR-130a-3p. To our knowledge, they have not been investigated in CMS before. Hsa-miR-205-5p was found to be a key regulator of VEGFA during cancer-related angiogenesis in hepatocellular carcinoma [42]. Since miRNAs are extremely stable even at room temperature and survive long freezing–thawing cycles without notable changes in expression [24], these small molecules have become more and more prominent research targets for human cancer detection and classification.

As shown in Figure 4, in all investigated miRNA expressions, the highest dCP values were found in the CIN3 group, and the dCP values were lower in the healthy control group (N). The dCP values for the HPV group were in between, except for miRNA 205-5p. The obvious tendency of the investigated miRNAs to differentiate between clinically relevant pathology (CIN3) and normal patients is displayed in Figure 2, Figure 3 and Figure 4. The corresponding *p* values did not reach statistical significance. Also, the observed tendency cannot be extended to the HPV group. The heterogeneous results of the HPV group are anticipated and probably related to the natural HPV cycle and latency. Prolonged high-risk HPV expression and positivity might result in high-grade lesions over time; this cannot be addressed because of the nonlongitudinal design of the study.

Previous studies have shown that miRNA profiling can offer a more accurate classification of human cancers compared to mRNA profiling. MiRNA profiling has also been proven to be useful in the diagnosis and prognosis of both solid cancers and haematological malignancies. This highlights the potential of miRNA analysis as a valuable tool in cancer research and clinical practice. [27,28,43,44,45,46,47,48,49,50,51]. Aberrant expression of miRNAs in different precursor lesions of cervical cancer has been investigated previously [52,53,54]. However, it is important to note that the results of these studies have not always been consistent, and there is still some ambiguity regarding the precise roles of miRNAs in supporting or suppressing carcinogenesis. This variability in findings may be attributed to differences in the sources of materials (such as tissue samples or cell lines) and methods used for miRNA analysis [52,53]. In a study conducted by Kawai et al., they identified four miRNAs (miR-126-3p, miR-20b-5p, miR-451a, and miR-144-3p) that were significantly upregulated in CIN 3 lesions. This was achieved using a miRNA microarray analysis of cervical mucus collected with a specialized cotton swab specifically designed for the experiment [53]. However, this collection method is not commonly used in regular clinical practice. The researchers reported an accuracy rate of 80% in detecting CIN 3+ lesions using this expensive and specialized method, regardless of the transformation zone. In our study, we aimed to make the collection of materials as noninvasive as possible, and we used simple and economical detection methods. This was done with the intention of considering the potential future use of the method in large-scale screening populations. By pursuing a less invasive and cost-effective approach, we aimed to develop a method that could be applicable in a wider clinical setting, facilitating broader access to early detection and screening for cervical cancer.

The presented study has some limitations that need to be addressed. The number of included patients is small compared to the amount of collected patient samples. As we are trying to fill a very specialized diagnostic gap by including only patients with a T3 transformation zone, we tried to keep the three different groups as clean as possible and excluded all patients with cervical dysplasia that was not confirmed as CIN3 and patients with discrepant results of cytology and histology. Due to the limited sensitivity of the diagnostic tools at hand and the natural cycle of HPV infection, it remains difficult. Latent HPV infections cannot be ruled out in the healthy subgroup (N).

The use of cervical mucosa causes a lot of variance. Within the menstrual cycle, its consistency changes immensely, which makes it difficult to obtain comparable amounts. Also, the endocervical swab often causes bleeding within the vulnerable transformation zone, which is, of course, also a potential source of bias. For future research, the use of serum or first-void urine are potential promising sources for the detection of miRNA that we will explore [26,55].

The data presented here can provide a basis for future research into a triage test for patients with a T3 transformation zone on the basis of commonly used clinical equipment and without the pain and discomfort associated with a colposcopic examination.

## 5. Conclusions

Our data indicate a tendency for differential expression of selected miRNAs in patients with high-grade cervical intraepithelial lesions (CIN3) and a T3 transformation zone compared to healthy controls. This is the first step towards finding a biomarker for cervical dysplasia in a patient collective in which the diagnostic gold standard tends to fail.

## Figures and Tables

**Figure 1 diagnostics-13-03599-f001:**
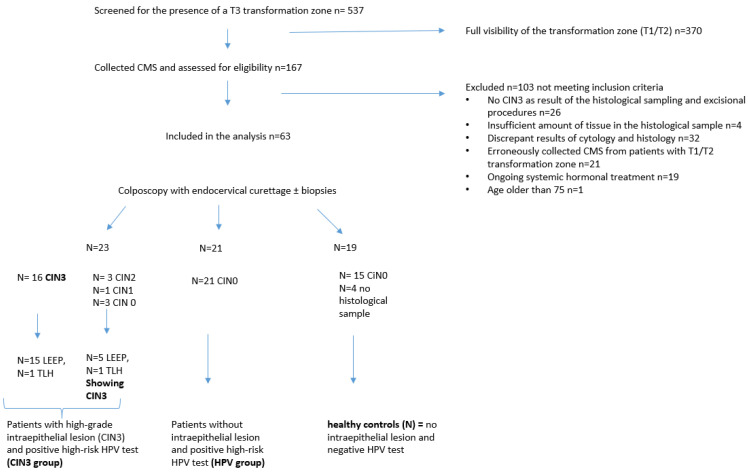
Patient flow chart.

**Figure 2 diagnostics-13-03599-f002:**
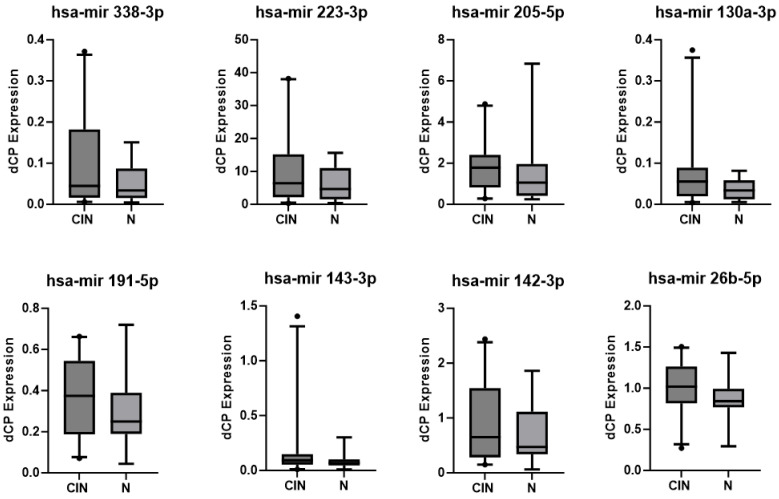
Boxplots showing the results of miRNA expression in patients with CIN3 (demonstrated as CIN here) and healthy individuals (N). The whiskers stand for the 5th and 95th percentiles of the data set. All other observed data points outside the boundary of the whiskers are plotted as outliers (dots).

**Figure 3 diagnostics-13-03599-f003:**
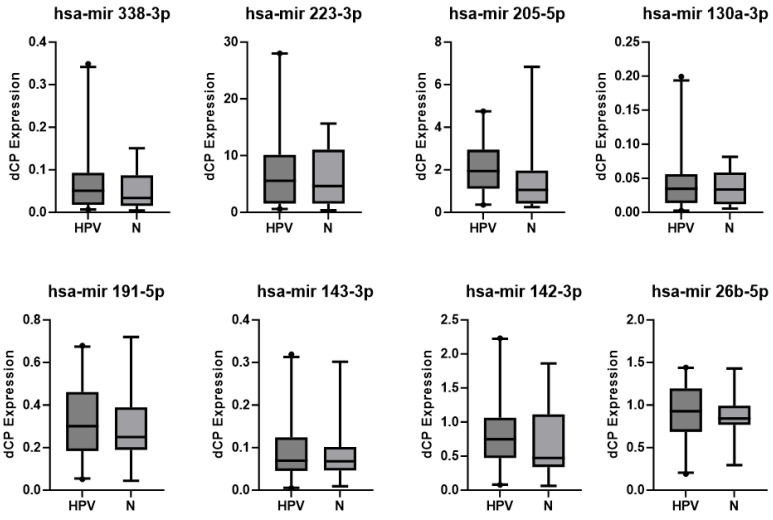
Boxplots showing the results of miRNA expression in patients with HPV and healthy individuals (N). The whiskers stand for the 5th and 95th percentiles of the data set. All other observed data points outside the boundary of the whiskers are plotted as outliers (dots). Long whiskers show the variability outside the upper and lower quartiles, which are indicated by the box. The sample median is indicated within the box.

**Figure 4 diagnostics-13-03599-f004:**
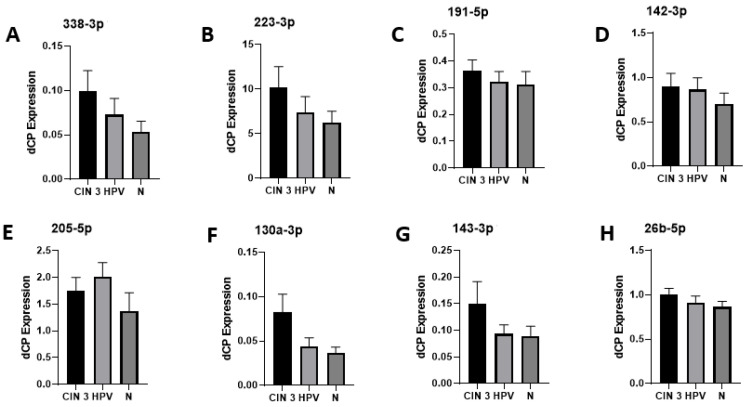
miRNA expression changes between patients with CIN 3(HSIL) and high-risk HPV infection (CIN3 group), patients with high-risk HPV infection without intraepithelial neoplasia (HPV group) and healthy individuals for expression of (**A**) hsa-miR-338-3p, (**B**) hsa-miR-223-3p, (**C**) hsa-miR-191-5p, (**D**) hsa-miR-142-3p, (**E**) hsa-miR-205-5p, (**F**) hsa-miR-130a-3p, (**G**) hsa-miR-143-3p, and (**H**) hsa-miR-26b-5p. dCP = difference in the crossing point, which describes the number of cycles at which the fluorescence exceeds the threshold divided by the SEM (standard error of the mean).

**Table 1 diagnostics-13-03599-t001:** Patient characteristics.

	HPV + CIN3 (CIN3-Group)	HPV Only (HPV Group)	Healthy Control Group (N)
Total number	23	21	19
Age (median)	42	51	46
Result of cytology upon referral			
PAP I (NILM) PAP IIp (ASC-US)/PAPIIID1 (LSIL) PAP IIIp (ASC-H) PAP IIID2 (HSIL) PAP IVap (HSIL) Unknown	0	21 (100%)	4 (21%)
	3 (13%)	0	6 (31.6%)
	4 (17.4%)	0	2 (10.5%)
	6 (26.1%)	0	2 (10.5%)
	9 (39.1%)	0	1 (5.3%)
	1 (4.3%)	0	4 (21%)
Patients with previous positive screening results	23 (100%)	21 (100%)	11 (57.9%)
Control cytology prior to colposcopy			
PAP I (NILM) PAP IIp (ASC-US)/PAPIIID1 (LSIL) PAP IIIp (ASC-H) PAP IIID2 (HSIL) PAP IVap (HSIL) Unknown	2 (8.7%)	21 (100%)	12 (63.2%)
	6 (26.1%)	0	7 (36.8%)
	2 (8.7%)	0	0
	4 (17.4%)	0	0
	9 (39.1%)	0	0
	0	0	0
HPV			
High-risk HPV Type 16/18 positive High-risk HPV Type 31/33/35/52/58/45 positive High-risk HPV Type 51/56/39/59 positive High-risk HPV Type 68/73/66 positive Both 16/18 and other positive Low-risk HPV positive	15 (65.2%)	8 (38.1%)	0
	12 (52.2%)	7 (33.3%)	0
	3 (13.0%)	3 (14.3%)	0
	0	3 (14.3%)	0
	6 (26.1%)	5 (23.8%)	0
	5 (21.7%)	4 (19.0%)	0

**Table 2 diagnostics-13-03599-t002:** Results of colposcopy.

	CIN3 Group	HPV Group	Control Group (N)
Minor changes	2 (8.7%)	9 (42.9%)	8 (42.1%)
Major changes	17 (73.9%)	0	0
normal	3 (13%)	12 (57.1%)	11(57.9%)
inadequate	1 (4.3%)	0	0
Endocervical curettage and Biopsies	19 (82.6%)	10 (47.6%)	8 (42.1%)
endocervical curettage only	4 (17.4%)	9 (42.9%)	7 (36.8%)
TZ 3 colposcopies illustrating ectocervical dysplasia	14 (60.9%)	0	0
No histological sample	0	0	4 (21.1%)

**Table 3 diagnostics-13-03599-t003:** Results of the Mann–Whitney *t*-test.

miRNA	*p*-Wert (CIN3 vs. N)
Hsa mir 338-3p	0.3261
Hsa mir 223-3p	0.3716
Hsa mir 205-5p	0.1712
Hsa mir 130a-3p	0.1367
Hsa mir191-5p	0.4379
Hsa mir 143-3p	0.2842
Hsa mir 142-3p	0.6090
Hsa mir 26b-5p	0.0893

**Table 4 diagnostics-13-03599-t004:** Results of the Mann–Whitney *t*-test. Significant values are indicated in bold.

miRNA	*p*-Wert (HPV vs. N)
Hsa mir 338-3p	0.6258
Hsa mir 223-3p	0.8875
Hsa mir 205-5p	0.0422
Hsa mir 130a-3p	0.9124
Hsa mir191-5p	0.6711
Hsa mir 143-3p	0.9374
Hsa mir 142-3p	0.3688
Hsa mir 26b-5p	0.6483

## Data Availability

The datasets generated during the current study are available from the corresponding author upon reasonable request.

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
