# Peer review of "A Glimmer of Hope for Patients with a T3 Transformation Zone: miRNAs Are Potential Biomarkers for Cervical Dysplasia"

_diagnostics, 2023, doi:10.3390/diagnostics13243599_

Round 1

Reviewer 1 Report

Comments and Suggestions for Authors

This pilot study assesses the potential use of miRNA’s in the triage of colposcopy patients with Type 3 (non-visible) cervical transformation zone. TZ3 is a constitutional finding, predominantly determined by hormones, obstetric history as well as the lack of estrogens, representing a common situation during menopause. TZ3 is related with controversies in colposcopy patient management, requiring skilled physician input and often multidisciplinary approach. With the ageing population globally, it is anticipated to represent a growing issue in the forthcoming decades.

The title “Glimmer of hope” in lay terms might attract some extra reads but is also somewhat sentimental.

Introduction: With a topic which mainly appeals to specialized clinicians, instead of repeating trivial knowledge on cervical HPV biology, authors might instead provide some insight on current guidelines issued by scientific societies on the management of TZ3; some recent references would be helpful. The authors correctly address anxiety issues related with expectant management TZ3 patients, but do not mention the excess unnecessary morbidity related with conization procedures & subsequent cervical stenosis; again references would be helpful.

Line 84: Even in expert hands colposcopy has suboptimal sensitivity & specificity, therefore it does not represent the “gold standard”; strangely, in line 95 the “gold standard” changes to endocervical curettage (!!), which also illustrates suboptimal performance characteristics (70%-81% sensitivity - line 264). LEEP/LLETZ most likely DOES represents the real gold standard, however it cannot be offered to every TZ3patient because of costs and the aforementioned subsequent psychosomatic morbidity.

Materials & Methods: State of the art assays and molecular diagnostics have been utilized. Since Seegene Anyplex II HPV kit can offer extended genotyping, it would be interesting to see a breakdown according to carcinogenicity potential (16/18-45/HR-other/Intermediate/low risk) in the results section, despite the small patient’s number.

The percentages of i) patients with previous positive/negative screening or genotyping HPV results, as well as ii) TZ3 colposcopies illustrating ectocervical dysplasia are not mentioned.

Lines 132-135: The authors state that “The indication for surgical therapy, loop electrosurgical excision procedure (LEEP) or total laparoscopic hysterectomy (TLH), was set according to the German S3 guideline for the prevention of cervical cancer”, however it is not completely clear (e.g. in a table or chart) how many patients received either procedure.

Results: Lines 179-180: the corresponding Bethesda (TBS) categorization should be quoted in brackets (like in Table 1), for readers unacquainted with the German categorization.

Lines 181-183: Regarding the CIN3 group, the authors state that “In 16 patients colposcopic led biopsy or endocervical curettage revealed CIN3 and in 7 patients it was found in the later performed loop excisional procedure of the cervix (LEEP)” Does this mean that only 7 LEEP’s have been performed or that all patients received LEEP or hysterectomy , but CIN3 histology was not corroborated? In the first case, how were the remainder CIN3 cases managed? In general the breakdown of the management lacks clarity; authors might consider providing a more comprehensive flowchart (Figure 1) encompassing patient management.

Lines 214-215: The heterogeneous results of the HPV group are somewhat anticipated and probably related with HPV natural cycle and latency. Prolonged HR-HPV expression and positivity might result in high grade lesions over time; this cannot be addressed because of the non-longitudinal design of the study.

Discussion: Again, the 1st paragraph should not repeat established knowledge on HPV primary screening but rather focus on management dilemmas of TZ3, especially in the context of excess colposcopy referrals anticipated in the first years of primary HPV screening implementation. TZ3 management requires vigilant management and this must be addressed with extra references.

Lines 254-255: The authors admit that “One of the major difficulties of this study was the assignment of patients to the different study groups” Indeed, while dealing with small samples it is understandable to pool non exactly identical procedures to achieve statistically significant results, however it might lead to potentially misleading results.

Lines 257-258: “Unfortunately, we cannot be 100% sure, if the patients in our healthy control group were actually completely healthy” Previous HPV screening/genotyping results could identify pts with latent infections.

Lines 292-293: “It remains unclear, whether in a larger collective of patient’s discrimination between productive and transforming HPV infection will be possible”. – Please clarify.

Line 312: “We tried to keep the three different groups as clean as possible” Discrimination of clear-cut subgroups from an HPV-biology standpoint is extremely difficult in this context; please refer to comment re Lines 254-255 as above.

English is fine; some minor polishing might be required.

Despite the aforementioned limitations, predominantly the small patient sample and the fluidity in assigning pt subgroups, this study’s results are indeed interesting in the context of a pilot research assessing the performance of a novel molecular triage biomarker panel in colposcopy cases illustrating obvious difficulties in management. For this reason I would like to see this study published soon.

Comments on the Quality of English Language

Minor language polishing might be needed.

Author Response

Dear Reviewer, please find the responses to your valuable comments in the attached document.

Thank you so much for your input.

Reviewer 2 Report

Comments and Suggestions for Authors

1. Excluding criteria: Chlamydia and other urogenital infections in time of investigation or history?

2. "...In some patients with multifocal lesions, more than one biopsy was taken..." How much standardly and what was the main target area?

3. Table 1. HPV + CIN3 Control cytology...only 60.9 percent?

4. Table 3. In the CIN3 group minor/major/normal changes, I calculated 95.6 percent (must be 100), like in the HPV and Control groups.

5. ...electrosurgical excision..., but laser surgery is much better.

6. Fig. 1. In 338-3p; 205-5p;130a-3p;143-36 high variance of data scatter. What about statistical reliability? The same is in Fig.3.

7. Since the samples are really small, maybe it would be better to use Shapiro-Wilk testing.

8. miRNA microarray analysis of cervical mucus - is it present in regular clinical practice?

9. Why was the age group over 40 chosen meaning that the menstrual cycle can consistently change and its important in this study?

Comments on the Quality of English Language

1. Excluding criteria: Chlamydia and other urogenital infections in time of investigation or history?

2. "...In some patients with multifocal lesions, more than one biopsy was taken..." How much standardly and what was the main target area?

3. Table 1. HPV + CIN3 Control cytology...only 60.9 percent?

4. Table 3. In the CIN3 group minor/major/normal changes, I calculated 95.6 percent (must be 100), like in the HPV and Control groups.

5. ...electrosurgical excision..., but laser surgery is much better.

6. Fig. 1. In 338-3p; 205-5p;130a-3p;143-36 high variance of data scatter. What about statistical reliability? The same is in Fig.3.

7. Since the samples are really small, maybe it would be better to use Shapiro-Wilk testing.

8. miRNA microarray analysis of cervical mucus - is it present in regular clinical practice?

9. Why was the age group over 40 chosen meaning that the menstrual cycle can consistently change and its important in this study.

Author Response

Dear Reviewer,

please find the response to your valuable comments in the attached document.

Thank you very much for your input.
